# ShapeEmbed: a self-supervised learning framework for biological shape analysis

## Abstract

The shape of objects is an important source of visual information in a wide range of applications. One of the core challenges of shape quantification is to ensure that the extracted measurements remain invariant to transformations that preserve an object's intrinsic geometry, such as changing its size, orientation, and position in the image. In this work, we introduce ShapeEmbed, a self-supervised representation learning framework designed to encode the outline of objects in 2D images into a shape descriptor that is invariant to translation, scaling, rotation, reflection, and outline point indexing. ShapeEmbed relies on a Euclidean distance matrix representation of the outline of input objects. Our approach overcomes the limitations of traditional shape descriptors while improving upon existing state-of-the-art autoencoder-based approaches. We demonstrate that the descriptors learned by our framework outperform their competitors in shape classification tasks on natural and microscopy images. Our framework is also generative, thus allowing for sampling and full reconstruction of 2D outlines from their latent feature vectors.

## 1. Introduction

The outline of objects in 2D images carry essential information about their shape. In natural images, humans are often able to recognize objects purely based on their silhouette without relying on texture or color (Wagemans et al., 2008). Interestingly, shape information is unaltered by many geometric operations such as similarity transformations (Dryden & Mardia, 2016) and is also unaffected by irrelevant and distracting imaging variables, such as lighting conditions or imaging setups. This is particularly relevant in biological imaging, where the shapes of living systems

extracted from microscopy images can reveal information about underlying biological processes, such as cell state or identity, across a wide range of imaging scales, settings, and modalities (Paluch & Heisenberg, 2009; Rangamani et al., 2013; Grosser et al., 2021; Zinchenko et al., 2023). All of these aspects make shape a highly desirable abstraction from pixel-intensity based images, enabling visualization, outlier detection, and unsupervised discovery of underlying patterns (Loo et al., 2007; Sailem et al., 2015).

The standard way of describing objects in 2D images is with binary segmentation masks, where pixels inside of an object's outline are set to $1$ and pixels outside to $0$. However, while such a representation is readily produced by segmentation algorithms and allows to abstract from lighting and imaging conditions, it is not invariant to transformations such as translation, rotation, reflection, and scaling. As such, the same object appearing twice in an image at a different location or orientation will yield segmentation masks that can only be recognized as equivalent after tedious processing. To circumvent this and preserve invariance with respect to similarity transformations, shape information is traditionally captured through statistics computed from the mask image, such as region properties (*e.g.*, area and curvature) or Fourier descriptors (Pincus & Theriot, 2007). Such methods, however, are averaging and condensing information by design, thus providing an incomplete description from which it is impossible to fully reconstruct the original outline in all of its details.

Representation learning has recently gained attention as a strategy utilizing autoencoders (Hinton & Salakhutdinov, 2006; Kingma & Welling, 2014) to derive descriptors that are able to capture all intricacies of object shapes while producing descriptors that are invariant to irrelevant geometric transformations. The vast majority of the methods proposed so far (Chan et al., 2020; Ruan & Murphy, 2019; Vadgama et al., 2022; 2023) aim to encode segmentation masks by relying on complex training strategies to ensure that the resulting latent code representations are geometrically-invariant.

Here, we introduce ShapeEmbed, a novel approach to extract shape descriptors relying on representation learning that leverages a simple architecture and training procedure to ensure invariance to translation, scaling, rotation, and reflection. Instead of directly encoding segmentation masks,

[1] Anonymous Institution, Anonymous City, Anonymous Region, Anonymous Country. Correspondence to: Anonymous Author <anon.email@domain.com>.

Preliminary work. Under review by the International Conference on Machine Learning (ICML). Do not distribute.

we propose to encode instead a distance matrix (Dokmanic et al., 2015) representation of object outlines. The distance matrix contains all pair-wise distances of the points on the outline of an object and is inherently invariant to translation and rotation. It also fully describes the outline and allows reconstructing it via multi-dimensional scaling (MDS) (Cox & Cox, 2000) without loss of information. On the other hand, distance matrices are not invariant to the indexation of points along the outline (*i.e.*,choice of origin and direction of travel). Different indexations however can be identified to result in elementary permutations of rows and columns in the matrix. Leveraging this property of distance matrices, we are able to implement invariance to indexation in the encoding step through a specific architecture of the encoder and the inclusion of a new loss function, leading to a latent descriptor of shape that is robust to all irrelevant geometric transformations.

Distance matrices have been used for a long time to characterize shapes and to compute shape dissimilarities without alignment (Hu et al., 2012; Konukoglu et al., 2012; Govek et al., 2023). While the use of pairwise point distances in these previous works is similar to what we propose, we do not use these point distances directly but instead as an input to a representation learning model that maps outlines to points in a latent shape descriptor space with generative properties. Our approach has similarities with Alphafold (Jumper et al., 2021), where distance matrices are used to describe the structure of proteins. However, as proteins are open linear structures with a clearly defined start and end point, the problem of indexation invariance encountered with closed outlines does not arise. We are thus, to the best of our knowledge, the first to overcome this issue and propose a framework to encode distance matrices of closed outlines within a VAE.

We evaluate our method by using a simple logistic regression classifier applied to the latent representation as a downstream shape classification task. We demonstrate that ShapeEmbed outperforms traditional statistics-based as well as learning-based methods on a range of different problems, including computer vision benchmarks and biological imaging datasets. Further quantitative exploration of the structure of our latent space indicates that its structure captures meaningful aspects of the shape of objects in images.

In summary, our contributions are as follows:

1. We introduce a self-supervised representation learning model that ingests distance matrices to learn shape descriptors that are by design insensitive to scaling, translation, rotation, reflection, and re-indexing.

2. We propose a novel indexation-invariant VAE architecture based on a padding operation in the encoder

operating jointly with a new loss function.

3. We show that our method outperforms the representation learning state-of-the-art and classical baselines on downstream shape classification tasks.

## 2. Related Work

We here review previous works on image-based shape quantification that are relevant to the approach we propose.

**Statistics-based methods** Shape quantification relying on summary statistics aims to assemble a large-enough collection of features, assuming that their ensemble provides a sufficiently complete description of the object's shape. The features themselves are handcrafted by design and most often consist of quantities such as area, perimeter, and curvature (van der Walt et al., 2014). Due to its simplicity and good empirical performance, this approach is overwhelmingly used in biological imaging (Bakal et al., 2007; Barker et al., 2022). Many summary statistics are inherently invariant to geometric transformation such as rotation and translation, but only partially capture shape information. As such, they are thus often unable to distinguish subtle shape differences.

**Decomposition methods** Decomposition methods seek to approximate an object's shape by a set of basis elements. The shape descriptor then corresponds to the coefficients of that approximation, and the original outline can be reconstructed as a weighted sum of the basis elements. The most common example of decomposition-based shape descriptors are the Elliptical Fourier Descriptors (EFD) (Persoon & Fu, 1977; Kuhl & Giardina, 1982). EFD are inherently invariant to similarity transformations, but often perform poorly in classification tasks as discriminative information tends to be hidden in noisy higher-order approximation coefficients.

**Learning-based methods** Following the success of autoencoders (Hinton & Salakhutdinov, 2006) and variational autoencoders (VAE, (Kingma & Welling, 2014)) for representation learning, self-supervised learning of shape descriptors directly from object masks appeared as a natural strategy to alleviate the shortcomings of classical methods. Methods have been proposed to encode images of 2D objects into a latent representation of the underlying object's shape (Chan et al., 2020; Zaritsky et al., 2021), but are often not invariant with respect to translation, scaling, and rotation. To mitigate this issue, a generic prealignment step can be carried out (Ruan & Murphy, 2019). However, as shown in (Burgess et al., 2024), it does not consistently produce good results.

A framework that employs invariant risk minimization to learn invariant shape descriptors was recently introduced in (Hossain et al., 2024). The approach focuses on capturing invariant features in latent shape spaces parameterized by

deformable transformations. While being robust to environmental variations, this method does not explicitly focus on achieving invariance to geometric transformations in the resulting shape representations and is heavily tailored to medical imaging data, with limited applicability to other types of images.

Recently, (Vadgama et al., 2022; 2023) introduced a VAE model trained to produce a latent space that explicitly disentangles a geometric shape descriptor from the orientation of the input object. The decoder network takes the orientation-invariant shape descriptor together with the orientation as input and is thus able to reconstruct the original mask. Both of these methods are superficially similar to ours in that they use a VAE to and achieve rotation invariance. However, while (Vadgama et al., 2022; 2023) explicitly estimate a rotation using their encoder network, our method bypasses this step by using the already rotation-invariant distance matrix representation as input to the encoder. As neither of these works evaluate their method on a downstream task and unfortunately do not provide a code repository, we were unable to include them in our results comparison.

Most closely related to our work is O2VAE (Burgess et al., 2024), a VAE model that encodes segmentation masks into an orientation-invariant latent code representation. The key idea of this approach is to rely on an encoder with rotation-equivariant convolutional layers (Weiler & Cesa, 2019) together with pooling to achieve invariance. In this pipeline, a realignment step is required during training to orient the input with its reconstruction. While O2VAE uses an elaborate special encoder to achieve rotation invariance, our method is inherently rotation-invariant due to its use of a distance matrix representation and only requires simple modifications to the VAE architecture to achieve indexation invariance.

## 3. Proposed Approach

ShapeEmbed extracts the outline of objects in 2D segmentation masks to construct a distance matrix representation that is then used to train a VAE model to learn a latent representation of shape. Thanks to a combination of the distance matrix properties and of the VAE model design, the resulting latent codes are invariant to translation, rotation, reflection, scaling, and point indexation (Figure 1). In the following, we describe ShapeEmbed step by step and discuss how we achieve these different types of invariance in our framework.

### 3.1. From Segmentation Masks to Distance Matrices

Starting with a 2D binary segmentation mask, we first interpolate the object outline with a parametric spline curve that we uniformly sample starting at an arbitrary position on the outline and going counterclockwise to yield a fixed number

$N$ of points $\mathbf{x}_i = (x_i, y_i)$. $N$ is a hyperparameter that we set to 64 by default, and that can be adjusted depending on the number of pixels composing the outlines of the considered objects. We then construct the corresponding $N \times N$ distance matrix $D$ with entries

$$d_{i,j} = |\mathbf{x}_i - \mathbf{x}_j|, \tag{1}$$

which is the Euclidean distance between points $\mathbf{x}_i$ and $\mathbf{x}_j$. Distance matrices are naturally invariant to translation and rotation and can be straightforwardly normalized to be made invariant to scaling upon division by the matrix norm, as demonstrated in Appendix A.1.

As they rely on points along the object outline, distance matrices are sensitive to the choice of origin (starting point) and direction of travel (clockwise or counterclockwise) on the outline, which impact the ordering of the matrix entries. Upon changing the starting point and/or direction of travel, the matrix entries will be shifted diagonally (change of origin) as well as horizontally and vertically mirrored (change of direction of travel), as illustrated in Figure 2.

More precisely, for a given distance matrix $D$, we denote the equivalent distance matrices obtained by choosing point number $k \in \{0, \ldots, N-1\}$ as origin and $o \in \{1, -1\}$ as direction of travel as $D^{k,o}$. This yields a total of $2N$ different equivalent matrices representing the same outline. The matrix entries are given by

$$d_{i,j}^{k,o} = d_{(io+k) \bmod N, \, (jo+k) \bmod N}, \tag{2}$$

where $d_{i,j}$ are the entries of the original distance matrix $D$, with the first point in the outline acting as origin.

We propose a minor modification to the encoder architecture in our VAE that makes it unable to distinguish between these re-indexations. Together with a modified loss function, our VAE is thus able to map all possible equivalent indexings of the outline to the same latent vector. Importantly, solving the indexation problem also grants our approach invariance to mirror reflection: assuming a fixed choice of origin and direction of travel, a mirror reflection of the outline will indeed correspond to a change of direction of travel, resulting in a distance matrix that is mirrored horizontally and vertically.

### 3.2. VAE Model with Custom Indexation Invariant Encoder

ShapeEmbed relies on a VAE model that encodes distance matrix inputs into a latent code representation that is invariant to irrelevant geometric transformations of the original outline.

Since distance matrices are 2D structures, they naturally lend themselves to being processed by powerful and established convolutional backbones developed for image

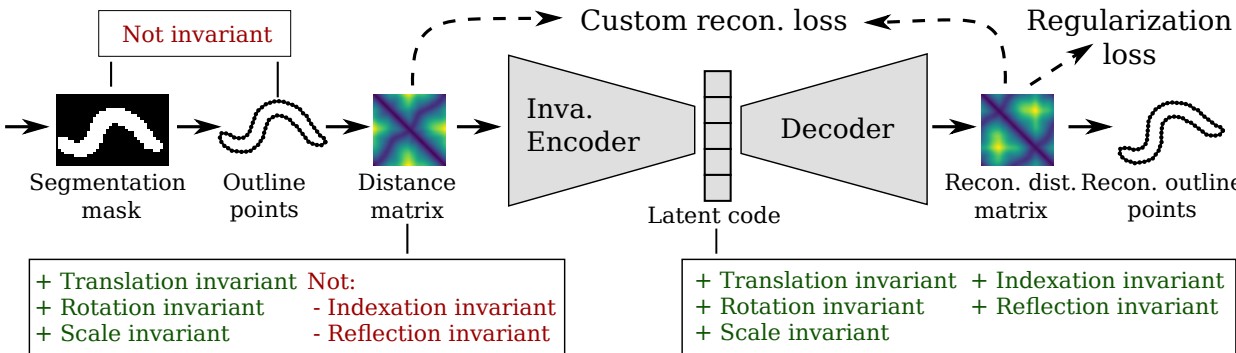

*Figure 1.* Overview of ShapeEmbed. ShapeEmbed converts the outline of an object from a 2D segmentation masks into a normalized distance matrix representation that is translation-, rotation-, and scale invariant. Relying on a VAE model, it then encodes distance matrices into a latent representation that adds indexation and reflection invariance. The resulting latent code forms a powerful shape descriptor that can be used for downstream tasks such as classification, and allows reconstructing the original outline, albeit arbitrarily indexed, rotated, translated, and reflected.

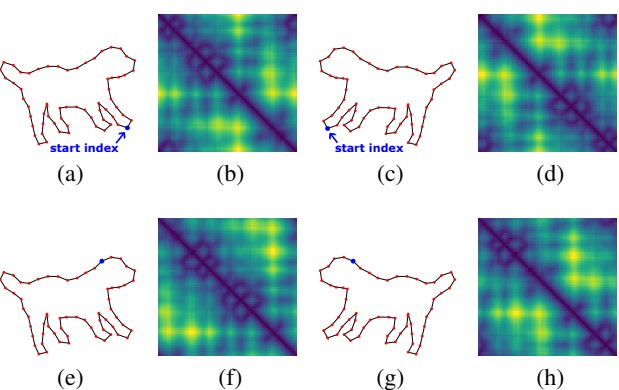

*Figure 2.* Effect of indexation changes on the distance matrix. An outline (a) and its corresponding distance matrix (b) obtained by traveling the outline counterclockwise from a given choice of origin (start index). Changing the direction of travel is equivalent to traveling through a mirror-reflected version of the outline in the counterclockwise direction (c) and yields a distance matrix that is mirrored horizontally and vertically (d). A different choice of origin (e) produces a diagonally-shifted version (f) of the original distance matrix (b). When combined (g), these operations produce a diagonally-shifted and mirrored version (h) of the original distance matrix (b).

data (Bengio et al., 2013). In our implementation, we thus use an encoder network based on the ResNet-18 architecture (He et al., 2016) that we mirror in the decoder path.

Remembering that our normalized distance matrices are invariant with respect to translation, rotation, and scaling, but not with respect to point indexation, we designed a novel indexation invariant encoder architecture to ensure that our latent codes only carry information about intrinsic shape.

As outlined in 3.1, different choices of origin on the out-

line result in distance matrices that are shifted diagonally. Conveniently, the convolutional layers in ResNet-18 are in principle shift equivariant, meaning that a shifted input will result in an identical but shifted output. Carefully considering boundary conditions, we propose to use circular padding (*i.e.*, padding by repeated tiling) in every convolutional layer, which directly corresponds to the modulo operation in 2. As a result, the convolutional layers are equivariant and produce equal but shifted outputs for all possible distance matrix indexations. We have to note that ResNet-18 does not exclusively use stacked convolutions, but also reduces size via stride and pooling. Strictly speaking, when convolutions are used within such architectures, the result is no longer truly shift equivariant or invariant (Rumberger et al., 2021). We however observe that, in practice, our architecture is sufficient to help prevent the latent codes from capturing indexation, as demonstrated in Section 4.

Our final encoder backbone is therefore a modified ResNet-18 where the standard convolutional and pooling operations are replaced with layers that incorporate circular padding. To make our encoder additionally invariant to the direction of travel of the outline (see Section 3.1), we process each matrix twice using the backbone, once in its original form and once horizontally and vertically mirrored. We then sum the two resulting output vectors to create an architecture that is unable to distinguish between a matrix and its mirrored version, rendering it invariant with respect to reflection.

### 3.3. Loss

**Indexation Invariant Reconstruction Loss.** Considering that our encoder is sufficiently invariant with respect to indexation, it follows that next to no information about the indexation of the outline is present in the latent code. This is a problem when computing the reconstruction loss:

as the same latent code could have been created by any shifted and mirrored version of the distance matrix, it is impossible to know which version of the matrix should be reconstructed to match the input - and as a matter of fact any of these alternative versions is correct as they all describe the same outline. To account for this ambiguity, we introduce a novel reconstruction loss that equally rewards all equivalent versions. To compute it, we generate all $2N$ alternative versions $D^{k,o}$ of the input distance matrix. We then define the reconstruction loss as

$$\mathcal{L}_{\text{rec}}(\hat{D}, D) = \min_{k \in \{0, \ldots, N-1\}, o \in \{-1, 1\}} \text{MSE}(\hat{D}, D^{k,o}), \quad (3)$$

where $\hat{D}$ is the decoded distance matrix (reconstruction), $D$ is the true distance matrix (input), $D^{k,o}$ is an alternatively indexed version of $D$ (see (2)), and $\text{MSE}(\cdot, \cdot)$ is the mean squared error over all matrix entries. This approach ensures that the decoder learns to reconstruct a version of the input distance matrix that minimizes the reconstruction error regardless of the choice of origin and direction of travel. This effectively removes the ambiguity without losing indexation invariance. By incorporating this loss into the training process, the model is encouraged to focus on the intrinsic geometric structure of the outlines rather than being sensitive to the arbitrary order of their points.

**Distance Matrix Regularization Losses.** We use several Euclidean distance matrix properties to regularize the learning process and encourage the decoder to produce a distance matrix-like output, leading to the formulation of three regularization terms.

First, as the distance from a point to itself is null, all entries in the leading diagonal of the distance matrix should be zero. This translates to

$$\mathcal{L}_{\text{diag}}(\hat{D}) = \frac{1}{N} \sum_{i=1}^{N} \hat{d}_{i,i}^2, \quad (4)$$

where $\hat{d}_{i,j}$ is the $i$-th entry in the diagonal of $\hat{D}$. Secondly, as the Euclidean distance is non-negative, all entries should be greater or equal than zero. This translates to

$$\mathcal{L}_{\text{non-neg}}(\hat{D}) = -\frac{1}{N^2} \sum_{i=1}^{N} \sum_{j=1}^{N} \min(\hat{d}_{i,j}, 0). \quad (5)$$

Third and finally, since the Euclidean distance is symmetric, the matrix should be symmetric too. This translates to

$$\mathcal{L}_{\text{sym}}(\hat{D}) = \text{MSE}\left(\hat{D}, \hat{D}^{\top}\right). \quad (6)$$

**Overall Loss.** Putting everything together, we use the following weighted sum as a loss to train our model:

$$\mathcal{L}_{\text{VAE}} = \mathcal{L}_{\text{rec}} + \beta \mathcal{L}_{\text{KL}} + \gamma \mathcal{L}_{\text{diag}} + \delta \mathcal{L}_{\text{non-neg}} + \epsilon \mathcal{L}_{\text{sym}}, \quad (7)$$

where $\mathcal{L}_{\text{KL}}$ is the classical Kullback-Leibler divergence loss term (Kingma & Welling, 2014), $\mathcal{L}_{\text{reco}}$ is our custom reconstruction loss (3), and $\beta$, $\gamma$, $\delta$, and $\epsilon$ are scalar hyperparameters. The hyperparameter $\beta$ allows tuning the model to focus more on feature extraction and reconstruction (smaller $\beta$) or on producing a smooth latent space that can be used in a generative context (larger $\beta$) (Higgins et al., 2017). We empirically set it to $10^{-10}$ by default, as this value was observed to balance accurate reconstructions and meaningful sampling in the latent space. The hyperparameters $\gamma$, $\delta$, and $\epsilon$ are all set by default to $10^{-5}$, which was empirically found through hyperparameter tuning.

### 3.4. Outline Reconstruction

Although we assess the latent representation learned by ShapeEmbed in downstream shape quantification tasks, it is useful to be able to reconstruct outlines from the latent codes for visualisation and quality control purposes. Outline points can be retrieved from a distance matrix using the MDS algorithm (Cox & Cox, 2000). However, in spite of the regularization terms presented in 3.3, the outputs of ShapeEmbed are neither truly symmetric nor have a leading diagonal composed of perfect zeros, and are therefore not true distance matrices. These deviations are fortunately typically negligible and within numerical error range, meaning that the leading diagonal values can be set to zero without significant loss of information. To enforce symmetry, we also take the average of the matrix and of its transpose as $\frac{1}{2}(\hat{D} + \hat{D}^{\top})$. This operation averages across the leading diagonal and is guaranteed to produce a symmetric matrix, thus allowing us to apply MDS. The algorithm is initialized with a random set of 2D points and iteratively updates them to minimize the difference between the entries of the distance matrix and the Euclidean distances between the points. MDS is guaranteed to converge, but not to a global minimum. It is also not guaranteed to converge to the same solution every time, but the solutions it recovers are all equivalent up to rotation, translation, and reflection, meaning that the resulting outline will be arbitrarily rotated, translated and reflected. Since the distance matrices inputted to the model are normalized for scale, the scaling factor must be carried over to the post-processing step and applied to the output distance matrix before MDS if one wants to recover the originally-sized outline.

## 4. Experiments

In this section, we review the datasets and evaluation metrics we use in our experiments, provide the implementation details of our method, present and discuss the relative performance of ShapeEmbed against relevant competitors, perform in-depth ablation studies to inspect the importance of the various invariance properties granted by

our framework, and finally demonstrate the added value of ShapeEmbed to identify subtle phenotypes in biological images. ShapeEmbed is implemented in Python and is available at `https://github.com/link_to_repo` (link to be added in camera-ready version). Further implementation details are provided in Appendix A.2.

### 4.1. Datasets

**MNIST.** The MNIST benchmark dataset (Deng, 2012) consists of grayscale images of handwritten digits from 0 to 9, with approximately $7,000$ images per class, amounting to a total of $70,000$ images.

**MPEG-7.** The MPEG-7 CE-Shape-1 Part B dataset(mpe) is a benchmark for shape matching and retrieval tasks. It consists of $1,400$ binary masks of objects belonging to 70 classes, with 20 images per class. Each class represents a distinct object category, such as different animals, tools, or symbols, designed to cover a range of shape variability.

**MEFs.** The Mouse Embryonic Fibroblast (MEFs, (Phillip et al., 2021)) dataset is a challenging biological imaging dataset containing 300 images of multiple cells distributed across three classes: circle-patterned, triangle-patterned, and control (non-patterned) surfaces, with 100 images per class. Although the original dataset includes two color channels corresponding to an actin and a nuclei stain, we here only use the actin channel as it captures whole cells. We segmented each individual objects in the images, leading to a total of $26,198$ masks distributed into $3,192$ cells in the control, $6,624$ cells in the triangle, and $6,565$ cells in the circle class, respectively.

**BBBC010.** The Broad Bioimage Benchmark Collection 10 (BBBC010, (Ljosa et al., 2012)) is a biological imaging dataset designed to test phenotypic profiling at the whole organism level. It contains a total of $1,407$ individual binary masks of *C. elegans* nematodes divided into a live and a dead class, each containing 768 and 639 individuals, respectively.

Additional details on the experimental settings for each of the considered dataset is provided in Appendix A.3.

### 4.2. Baselines and Evaluation Strategy

We compare the performance of ShapeEmbed for shape classification against two classical shape analysis baselines (Elliptical Fourier Descriptors (Persoon & Fu, 1977) and Region Properties (van der Walt et al., 2014)) and its main representation learning-based competitor (O2-VAE (Burgess et al., 2024)). We use 19 Region Properties features that pertain to shape and calculate Fourier Descriptors up to the 30th order, resulting into a vector of 120 coefficients per object. Additional details on the implementation of these two methods are provided in Appendix A.4. For O2VAE, we use the native implementation provided in (Burgess et al., 2024),

*Table 1.* Classification performance (F1-score) of different shape descriptors on biological imaging datasets. Higher values indicate better performance.

| METHOD | MNIST | MPEG-7 |
|---|---|---|
| REGION PROPERTIES | $0.809 \pm 0.003$ | $0.701 \pm 0.014$ |
| EFD | $0.623 \pm 0.013$ | $0.079 \pm 0.008$ |
| O2VAE | $0.855 \pm 0.007$ | $0.629 \pm 0.053$ |
| **SHAPEEMBED** | $\mathbf{0.963 \pm 0.007}$ | $\mathbf{0.751 \pm 0.024}$ |

running their model with the recommended hyperparameters to ensure consistency and fairness with the published setup. While O2VAE can incorporate both shape and texture information, we here use binary masks as inputs to specifically focus on shape in our comparison.

To quantitatively evaluate the quality of different shape descriptors, we rely on a downstream classification task. We train a logistic regression classifier (Bisong, 2019) following a 5-fold cross-validation strategy, and report the mean and standard deviation of the F1-score as a performance metric. The F1-score balances precision and recall and thus provides a reliable measure of performance across the considered datasets (Ye et al., 2012), with higher F1-score indicating better performance.

### 4.3. Benchmarking

We quantitatively evaluate the performance of region properties, EFD, O2VAE, and ShapeEmbed on the MNIST and MPEG-7 datasets. We highlight in Table 1 the superior performance of ShapeEmbed over both the classical baselines and its representation learning competitor. We additionally report a different metric for the same experiment in Appendix A.5, which leads to the same conclusion. We stress that these experiments are not meant to push the state-of-the-art in MNIST classification, but instead to evaluate the information content of the shape representation learned by the different methods we consider.

### 4.4. Ablation Studies

The MNIST and MPEG7 datasets, in their original form, consist of objects that all have roughly the same size and that have been aligned and centered. To assess the practical merit of the various invariances granted by ShapeEmbed, we constructed modified versions of the MNIST and MPEG-7 datasets that incorporates size variability through random object scaling (referred to as Scaled MNIST and Scaled MPEG-7), as well as positional and rotational variability through random object translation and rotation (referred to as Rand MNIST and Rand MPEG-7). As a result, objects in these modified datasets neither appear centered nor aligned in the images and exhibit a wide range of different sizes.

**Scaling and Indexation Invariance.** We evaluate how im-

*Table 2.* Effect of normalization and indexation invariance on classification performance (F1-score) considering randomly scaled versions of MNIST and MPEG-7. "None" indicates no indexation and no normalization invariance.

| METHOD | SCALED MNIST | SCALED MPEG7 |
|---|---|---|
| NONE | $0.865 \pm 0.005$ | $0.238 \pm 0.025$ |
| NO INDEXATION INV | $0.884 \pm 0.012$ | $0.588 \pm 0.071$ |
| NO NORMALIZATION | $0.910 \pm 0.006$ | $0.415 \pm 0.018$ |
| **SHAPEEMBED** | **$0.948 \pm 0.004$** | **$0.699 \pm 0.085$** |

portant the normalization step and the various modifications implemented in the VAE to achieve indexation invariance are in the model's ability to maintain performance under varying object sizes and choices of origin. To assess the effect of our modified encoder and custom indexation invariant reconstruction loss, we created a modified version of ShapeEmbed in which the circular padding mechanism is replaced by a constant padding of $1$ and where the indexation invariant reconstruction loss (3) is substituted with the standard MSE reconstruction loss. To evaluate the effect of normalization, we simply skipped it and retained the original, non-normalized distance matrices. We report F1-scores on the Scaled MNIST and Scaled MPEG-7 datasets in Table 2. We observe that removing indexation invariance results in a drop of $7.24\%$ in performance on MNIST, while skipping the normalization step reduces performance by $4.18\%$ on that same dataset. Even more drastic performance drops can be observed on Scaled MPEG-7. These results illustrate that, when ShapeEmbed does not include scaling and indexation invariance, it captures features in the latent space that are irrelevant to intrinsic shape information and therefore interfere with downstream tasks.

**Rotation and Translation Invariance.** We test the robustness of our model to positional and orientation variations, which are frequently encountered in real-world data. Unlike scaling and indexation invariance, which are explicitly enforced in the model, rotation and translation invariance are inherent to the distance matrix representation we use in ShapeEmbed. Ablating the distance matrix representation thus results in encoding the image mask directly with a vanilla VAE model, that naturally doesn't have any mechanism to implement rotation and translation invariance. For the sake of completeness, we also include the performance of O2VAE as a reference, as it partially addresses rotation and translation invariance but still uses masks as input. The results reported in Table 3 illustrate the positive impact of the distance matrix representation. On both the Rand MNIST and the Rand MPEG-7 datasets, ShapeEmbed scores higher than any of the considered alternatives. The gap in performance between ShapeEmbed and the other considered approaches highlights the difficulty of extracting relevant shape features in the absence of explicit translation and rotation invariance in a dataset that exhibits great vari-

*Table 3.* Effect of the input representation (image masks VS distance matrices) on classification performance (F1-score) for randomly translated and rotated versions of MNIST and MPEG-7.

| METHOD | RAND MNIST | RAND MPEG7 |
|---|---|---|
| VANILLA VAE | $0.382 \pm 0.013$ | $0.042 \pm 0.021$ |
| O2VAE | $0.658 \pm 0.008$ | $0.102 \pm 0.023$ |
| **SHAPEEMBED** | **$0.846 \pm 0.011$** | **$0.656 \pm 0.052$** |

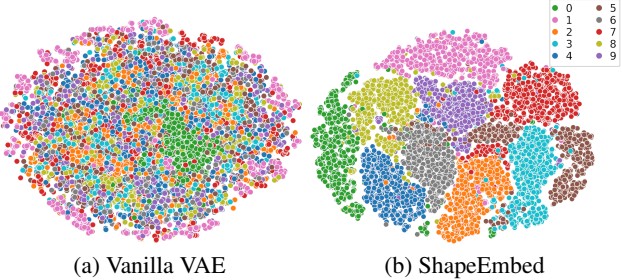

(a) Vanilla VAE          (b) ShapeEmbed

*Figure 3.* Projection (t-SNE) of the Rand MNIST latent space. (a) The latent representation learned by a vanilla VAE on a randomly rotated and translated version of MNIST does not exhibit any noticeable structure and class separation. In contrast, (b) the latent representation learned by ShapeEmbed, which ignores orientation and position, recovers clusters of data points that match their underlying class.

ability in object orientation and position, and demonstrates the value of the distance matrix representation.

To further qualitatively explore the effect of rotation and scaling invariance on the learned representation, we generated 2D projections of the latent space learned by the vanilla VAE and by ShapeEmbed relying on the t-SNE (van der Maaten & Hinton, 2008) dimensionality reduction technique. We display the t-SNE projections of the Rand MNIST latent space in Figure 3, where individual data points are colored according to the class label of their original input image. We observe that the latent representation learned by the vanilla VAE is randomly structured and does not allow resolving individual classes. The latent representation learned by ShapeEmbed, however, aggregates data points with similar class labels together, as one would expect the vanilla VAE to behave on the standard MNIST dataset composed of pre-aligned and centered objects. The t-SNE algorithm is used with a random seed of $42$ and a perplexity of $5$, which are commonly-used default parameters.

**Further Ablation Studies.** We experimentally explore two more ablation studies in our Supplementary Material: the added value of relying on the VAE latent codes as opposed to using distance matrices directly as shape descriptors (Appendix A.6), and the effect of the distance matrix regularization loss terms (Appendix A.7).

*Table 4.* Classification performance (F1-score) of different shape descriptors on biological imaging datasets.

| METHOD | MEF | BBBC010 |
|---|---|---|
| REGION PROPERTIES | $0.722 \pm 0.006$ | $0.821 \pm 0.002$ |
| EFD | $0.327 \pm 0.047$ | $0.523 \pm 0.001$ |
| O2VAE | $0.521 \pm 0.015$ | $0.659 \pm 0.076$ |
| SHAPEEMBED | $0.670 \pm 0.009$ | $0.837 \pm 0.035$ |
| SHAPEEMBED + SIZE | $\mathbf{0.745 \pm 0.021}$ | $\mathbf{0.859 \pm 0.007}$ |

### 4.5. Application to Biological Imaging

One of the main motivations for this work is its potential application to biological imaging. Shape analysis in biological images is particularly challenging as objects in these datasets typically appear unaligned, not centered, and may exhibit extensive size variations. Additionally, shape differences in biological systems often appear as subtle changes, the magnitude and nature of which is typically not known a priori. For these reasons, methods capable of quantitatively describing object outlines that are invariant to position and rotation, while remaining powerful enough to capture minute differences in shape are of strong interest. In biological imaging, size invariance may either be a crucial or entirely irrelevant feature depending on the context. It is necessary when size differences arise from imaging conditions (such as varying magnifications) but undesired when size differences are biologically meaningful (such as varying growth rate). While our framework is inherently scale-invariant, size can be retained as an additional feature by saving the norm of the distance matrix prior to normalization and can be added back in downstream tasks. We assessed the value of ShapeEmbed, with and without including size information, on biological imaging datasets at the organism (BBBC010) and cellular (MEF) scales and report performance against region properties, EFD, and O2VAE in Table 4. When adding back object size as an extra feature, ShapeEmbed consistently outperforms other considered methods. As objects in the MEF dataset exhibit experimentally-induced size differences between classes in addition to true shape variations, summary statistics, which include size-related metrics (such as the area), perform exceptionally well and better than the scale-invariant version of ShapeEmbed on this dataset. This observation highlights the importance of offering a flexible way to handle size information that can adapt to the biological question considered. In Appendix A.8, we additionally report a different metric for these experiments that leads to the same conclusion and also explore the generative properties of our model.

Further to quantitative classification results, we also qualitatively explore the latent space learned by ShapeEmbed on BBBC010 through the 2D t-SNE projection displayed in Figure 4 and obtained with the same parameters as Figure 3. Individual data points are colored according to the class

label of their original input image, which is either dead or alive. In BBBC010, labels have been derived from experimental conditions (whether the sample has been treated by a lethal substance or not). When dead, *C. elegans* nematodes straighten to look like a rod, while they swim sinusoidally and curve when alive. Upon inspection of the structure of the latent space learned by ShapeEmbed, we discover that several of the "misclassified" data points actually correspond to mislabeled individuals that are either alive despite having been treated, or dead despite being untreated. This interesting finding highlight the value of ShapeEmbed as a method to explore and discover shape variations in a fully unsupervised manner that allows uncovering subtle variations in biological experiments.

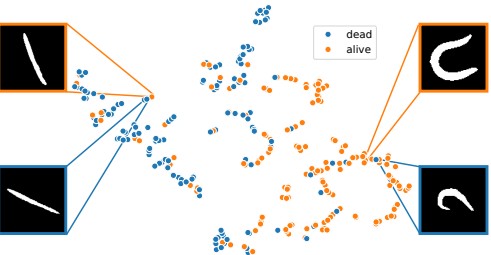

*Figure 4.* Projection (t-SNE) of the BBBC010 latent space learned by ShapeEmbed. Data points appear to group according to their corresponding classes, namely dead (straight rods) and live (curved worm) nematodes. A closer inspection of data points that seem to be misplaced reveals that their associated class label does not reflect their actual shape.

## 5. Conclusion

We introduced ShapeEmbed, an original self-supervised representation learning framework based on a custom VAE that can, from the image mask output of any segmentation algorithm, extract a latent representation of shape that is agnostic to position, size, orientation, and reflection. The key ideas behind our method are the use of distance matrices to encode the outline of objects, the implementation of simple but essential modifications to the encoder path of our VAE, and the use of novel loss terms. In our experiments, we demonstrated the superior performance of ShapeEmbed over existing methods for shape quantification over a range of natural and biological images. We also highlighted that ShapeEmbed is able to capture variability both across and within experimental conditions in biological images. We expect ShapeEmbed to be of valuable use for the unbiased exploration of shape variation in image datasets, and expect it to be most impactful in biological imaging where the size, orientation, and position of objects are highly unpredictible and shape differences are subtle. Although ShapeEmbed is currently limited to 2D images, it could serve as the basis for a 3D extension to be explored in future work.

## Impact Statement

This paper presents work whose goal is to advance the field of Machine Learning. There are many potential societal consequences of our work, none which we feel must be specifically highlighted here.

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
