# OpenReview forum: "ShapeEmbed: a self-supervised learning framework for shape quantification"
_ICML.cc/2025/Conference — Submitted to ICML 2025_

### Official Review · Reviewer_MfnJ · 2025-03-10

**Overall Recommendation:** 2

**Summary:**

This paper presents a self-supervised method for learning object shape, given a binary segmentation mask, that by construction is invariant to translation/scaling/rotation/reflection/outline-point-indexing.  This method, ShapeEmbed, consists of extracting a normalized distance matrix from points sampled along the outline of the object, and using this as input to a VAE, where the encoder is modified to use circular padding for convolutions, maintaining equivariance to shifts in the choice of origin for the point indexing.  Additionally, several loss terms are added - reconstruction loss of the distance matrix, and regularization terms enforcing a valid reconstructed distance matrix.

The proposed method is tested on several datasets - MNIST, shape matching, and two biological datasets.  Results show improvements over alternative baselines, and ablation studies demonstrate the contribution of individual components such as the circular padding and index-invariant loss.

## update after rebuttal

After reading the rebuttal and other reviews/discussions, I will maintain my original rating.  The data sets that were studied, combined with the need for the correct foreground segmentation to be provided, feel more like an initial proof of concept.  In my view the paper would be greatly strengthened by the inclusion of the type of analysis alluded to in the rebuttal, to validate the real-world applicability of the proposed method.

**Claims And Evidence:**

Yes, the experiment findings support the claims made regarding the properties and benefits of the proposed method.

**Essential References Not Discussed:**

n/a

**Experimental Designs Or Analyses:**

Useful ablation studies provided, in paper and supplementary pdf.

**Methods And Evaluation Criteria:**

I'd consider the experiment results on the studied datasets to be "proof of principle" - useful as an initial test of the proposed method, but not at the level of an dataset demonstrating the utility of the method on a real application.  For instance, it seems that if one were interested in doing classification on BBBC010, the attainable accuracy is 94% (Wählby et al., Nat Meth, 2012).

The studied datasets seem to be both simple in terms of the structure of the shapes, and in terms of the foreground segmentation being provided.  I think the paper would be greatly strengthened by showing an application on a more complex dataset where the segmentation was automatically inferred, and where the shape component gave some improvement to the overall performance as compared with what could be obtained without explicitly considering shape in this manner.

**Other Comments Or Suggestions:**

Could be useful to provide sample image/shape from MEF dataset, seems to be only one that is omitted as far as visualizations.

**Other Strengths And Weaknesses:**

n/a

**Questions For Authors:**

As discussed above, my key concern is with benefit of the proposed method to real world applications.  This is somewhat discussed in the penultimate sentence of the paper, but can the authors describe in more specifics how they see this method being applied to specific biological imaging datasets?

**Relation To Broader Scientific Literature:**

As noted above, one potential weakness of the proposed method is that it currently seems to only be tested on cases where the correct foreground segmentation is provided.  In practice, I imagine the segmentation problem itself will often be difficult and give noisy results.  One potential interesting avenue would be whether this method could be leveraged to quality of automatic segmentation, by providing a learned prior over the shape distribution for the given class of images.

**Theoretical Claims:**

Checked over the discussion of the construction of the method/encoder and loss to verify invariance claims.

---

> ### Author Rebuttal · Authors · 2025-04-01
>
> # Envisioned use of the method in biological imaging
> Morphological features extracted from 2D images serve as phenotypic fingerprints to reveal cell identity, cell states, and response to chemical treatments (see 10.1038/s41592-024-02241-6 for a recent example). Shape, as captured in 2D contours, is one of the most information-rich phenotypic characteristics and provides insights into a range of biological phenomena. Many methods have been proposed to characterize 2D shape in biological imaging (see 10.1111/j.1365-2818.2007.01799.x for a review) and unsupervised methods that include geometric invariances have recently gained traction as illustrated by O2VAE, our main competitor (10.1038/s41467-024-45362-4). This can be explained by the following: as an exploratory science, biology operates without knowing what to look for a priori, making unbiased data exploration invaluable. While experiments aim to uncover "biological labels" (e.g., cell type, cell state), living systems don't come with annotations and researchers only have access to "experimental labels" (e.g., treated vs untreated samples). Using these experimental labels as proxies for the underlying biological labels is inherently problematic due to individual variability: two samples treated identically may respond differently because of natural variations. Self-supervised approaches are especially valuable as they enable the investigation of biological labels independently of experimental categories. We demonstrate one such use-case in the BBBC010 dataset, where ShapeEmbed identifies distinct shape populations of C. elegans nematodes without supervision and reveals instances where experimental labels do not align with biological reality (Figure 4).
>
> Having methods that allow such an unbiased exploration of the distribution of biological shapes can be valuable in many settings, from assessing the efficacy of drug treatments to analyzing biopsies, where cell type identification must ideally be carried out without prior knowledge or potentially biased manual annotation. Although we cannot provide specific details for confidentiality reasons, we are already using ShapeEmbed in such contexts in ongoing collaborations with experimental colleagues.
>
> # Complexity of the biological datasets considered
> The biological datasets we use in our experiments were chosen to 1) provide results on a relatively simple and well-characterized biological benchmark (BBBC010), and 2) demonstrate performance on a harder, real-life example where the shape component is known to be essential (MEF).
>
> BBBC010 is indeed not a “pure” shape dataset. Although the authors report a classification accuracy of 97% and a precision of 83% (see 10.1038/nmeth.1984, note that 94% accuracy is for object detection, not classification), this comparatively better performance over ShapeEmbed (87% accuracy and 87% precision, see response to Reviewer HCmb) is likely due to the use of intensity and texture features. We do not use BBBC010 to demonstrate that we provide the best results when it comes to classification, but instead to show that we can learn, without any supervision, a good representation of biological shape in a way that allows unbiased exploration of the data. This also motivates not comparing our results with those reported by Wählby et al., but with a corresponding shape-only baseline (called Region Properties in Table 5). Further investigation of the representation space (Figure 4) illustrates that classification performance alone is not a good indicator of the ability to distinguish between biological states in this dataset, as some nematodes labeled live appear to be dead and vice-versa.
>
> The MEF dataset contains images of cells that were cultured on fibronectin micropattern surfaces to enforce cell shape constraints (as described in 10.1242/jcs.091231). This real use-case illustrates the challenge of untangling individual biological variability vs experimental variability and has been used in recent papers proposing unsupervised frameworks for biological shape analysis (10.1038/s41596-020-00432-x, 10.1038/s41467-024-45362-4). We regret not including visual examples from the MEF dataset illustrating the complexity of the data and would like to do so in the supplementary material of the final version. Until then, we refer to the original publication to get a better sense of the data complexity.
>
> # Other remarks
> We indeed rely on existing segmentation masks as we focus on the task of learning a representation of shape information. We agree with the idea of using ShapeEmbed to construct a learned shape prior to help with segmentation, which could either be pre-trained on unpaired segmentations or learned on-the-fly together with a future segmentation architecture to improve quality. Both approaches would be exciting future work but are beyond the scope of this paper.

---

### Official Review · Reviewer_C6GA · 2025-03-14

**Overall Recommendation:** 2

**Summary:**

Authors introduce a network for 2D shape analysis (silhouettes of objects in images). It works as follows: shape outline is interpolated via spline curve with fixed number of points N across all data samples; pairwise distance matrix is constructed based on those points and normalized to unit Frobenius norm; resulting adjacency matrix is treated as image and passed to convolutional auto-encoder. Proposed approach is translation/scaling/rotation invariant by design. Re-indexing invariance (distance matrix depends on starting point choice) is achieved via loss term that evaluates all valid perturbations of distance matrix. Method shows very strong performance compared to baselines across a diverse set of data.

POST-REBUTTAL UPDATE

I have read other reviews and agree with them on the following points:

+ that method also might not be able to handle noise in segmentation well and noise in segmentation can produce non simply connected shapes. To me, it is another practical concern about application of proposed niche method.
+ method seem to work really well for particular niche of biological data (even without comparison to PointNets and subsequent works).

To me, final decision boils down to potential changes we think are necessary for the paper to be accepted. Below are things that I think need to be changed for the paper to be accepted:

+ Positioning right now is too broad for actual contribution. I think it should be named "ContourEmbded:... " instead of "ShapeEmbed". In the rebuttal authors suggested the following change: “ShapeEmbed: a self supervised learning framework for 2D contour quantification”.  For me, it is too minor -- this paper is not about shape embeddings.
+ All discussion about method assumptions: simply connected shapes, non-noisy segmentation of silhouettes (?) should be added with examples (supplement is okay).
+ Additional evaluation should be added to paper or supplement as well.

For me those changes are too major to be accepted without revision, so I keep my rating.

**Claims And Evidence:**

- Proposed shape descriptors are invariant to scaling, translation, rotation, reflection and re-indexing (theoretical; see corresponding section).
- Proposed shape descriptors significantly outperform baselines (O2VAE, EFD - Elliptical Fourier Descriptors) for 2D shape classification tasks on diverse set of data: handwritten digits(MNIST); general shapes (MPEG-7); cell data (MEFs); nematodes data (BBBC010).
- Proposed method can be used in a generative setting to sample 2D outlines from latent feature vectors which is demonstrated qualitatively in supplement.

**Essential References Not Discussed:**

Paper largely ignores point based networks that can be leveraged for 2D shape analysis: 2D shapes can be represented as 2D point clouds and then processed via point networks (original PointNet paper has experiments on MNIST, for example). Point networks are usually indexing invariant by design; translation/scaling invariance is achieved by normalization; and rotation invariance is achieved by certain architectural modifications (see some references below):

[1] Li X, Li R, Chen G, Fu CW, Cohen-Or D, Heng PA. A rotation-invariant framework for deep point cloud analysis. IEEE transactions on visualization and computer graphics. 2021 Jun 25;28(12):4503-14.

[2] Zhang Z, Hua BS, Yeung SK. Riconv++: Effective rotation invariant convolutions for 3d point clouds deep learning. International Journal of Computer Vision. 2022 May;130(5):1228-43.

[3] Li F, Fujiwara K, Okura F, Matsushita Y. A closer look at rotation-invariant deep point cloud analysis. InProceedings of the IEEE/CVF International Conference on Computer Vision 2021 (pp. 16218-16227).

Paper also ignores a large body of work on shape descriptors (see reference below and follow-up works).

[1] Osada R, Funkhouser T, Chazelle B, Dobkin D. Shape distributions. ACM Transactions on Graphics (TOG). 2002 Oct 1;21(4):807-32.

**Experimental Designs Or Analyses:**

- Experimental design for classification is solid for in-distribution setting but does not show how well the method generalizes across datasets.
- Generative experiments design is lacking because evaluation is only qualitative.
- Choice of baselines is very limited and ignores a large body of work on rotation-invariant point networks that can be utilized for this problem.

**Methods And Evaluation Criteria:**

- Classification methods are evaluated via F-Score and log-loss (in supplement).
- For all datasets authors report metrics based on 5-fold cross-validation: mean and std across folds.
- Generative models are only evaluated qualitatively.

I find evaluation limited, especially for biological data. F-score is a good balanced measure for overall performance and results are convincing in that regard. However, additional metrics (e.g. precision, recall) might be helpful to better understand how exactly the proposed method outperforms baselines. I recommend including them in supplement.

**Other Comments Or Suggestions:**

- Legends in some figures (Fig. 3, Fig. 4) are hard to read. I recommend larger font for them.

**Other Strengths And Weaknesses:**

+ Empirical results are very strong. If combined with proper discussion of method assumptions and limitations, this might be useful work for practitioners.

**Questions For Authors:**

My decision is mostly based on re-indexing theoretical issues and limited comparison to point-based approaches. Can authors kindly clarify the following:

1) Does the method assume that 2D shapes are simply-connected? If yes, how does the method extend to non-simply connected shapes?
2) If the method does not assume simply connected 2D shapes, what is wrong with examples that I have provided in theoretical claims (O- and 8-shapes; touching circles)?

3) If the method assumes that shapes are simply connected, how often this is observed in evaluated data?

4) Are there any insights how the proposed method would work in comparison with rotation-invariant 2D point networks?

**Relation To Broader Scientific Literature:**

Proposed method introduces 2D shape surface representation based on the distance matrix of silhouette points and convolutional network that processes distance matrix as image.  This has following relations:


- The fact that 3D (and 2D shapes) can be thoroughly characterized via distribution of pairwise distances of surface points is well known in the 3D vision/graphics community (see essential references).
- Shape analysis in the form of surface point clouds is also a well-studied topic in the 3D vision community and can be applied for this problem as well (see essential references).

**Theoretical Claims:**

- Proposed method by design is invariant to translation, rotation and reflection since it relies on distance matrix of shape silhouette approximation. Scaling invariance is achieved by normalization of the distance matrix by matrix norm. This claim sounds solid to me.
- Authors also claim re-indexing invariance (method depends on order of rows/columns of distance matrices). This is achieved by loss that takes into account all possible distance matrices based on choice of initial point and orientation of the silhouette (overall 2N variants). Thus, this approach is not truly invariant to re-indexing but rather it is enforced via loss. This claim is justified empirically via ablation of this part of the loss (Table 2).

I find the last claim theoretically problematic. To me, it looks like the authors assume that all 2D shapes are simply connected when they derive 2N variants. For non-simply connected shapes this does not seem to hold. Let’s consider 2D donut (O-shape). Let’s assume that external and internal contours have N/2 points each (overall N). It means that indexing of internal and external contours have N variants each. But their interactions have N^2 variants (each external contour indexing can be matched with any internal contour indexing)! And if we assume 2D double donuts (8-shape) with N/3 points per contour, the number of variants becomes (2N/3)^3. I think the proposed method still works because most of the shapes in data are simply connected but this theoretical issue requires clear discussion in the paper and, ideally, ablation.

Another theoretical issue is related to contour traversal. To me, it looks like authors assume no loops in contour. Let’s consider two circles that touch at a single point. Authors assume 2 possible orientations of such a contour but in fact there are four: 2 traversals of one circle until we hit second and after that there are also two traversals. I consider this example to be more rare compared to non simply-connected 2D shapes but this still needs to be discussed.

---

> ### Author Rebuttal · Authors · 2025-04-01
>
> # ShapeEmbed assumptions on 2D shapes connectedness
>
> We thank you for your insightful questions and hereafter answer them one by one.
>
> **Does the method assume that 2D shapes are simply-connected?** Yes, ShapeEmbed operates with contours that are simply connected and described by a sequence of ordered points (i.e., the contour can be recreated by linking points that follow each other). If accepted, we will adjust the wording in the method description to make this clearer.
>
> **If yes, how does the method extend to non-simply connected shapes?** Our model architecture and loss rely on the fact that points in simply-connected contours can be unambiguously ordered (up to the choice of origin and direction of travel) to learn a representation that ignores reparameterization. For non-simply connected shapes, there is no obvious way to “concatenate” them into a single distance matrix as you rightfully pointed out. Dealing with this would require a model that is invariant to point ordering altogether (similar to point clouds). We are interested to explore this in future work, but it goes beyond the scope of ShapeEmbed.
>
> **If the method does not assume simply connected 2D shapes, what is wrong with examples that I have provided in theoretical claims (O- and 8-shapes; touching circles)?** Although ShapeEmbed assumes simply-connected 2D shapes as described above, it can still handle 0- and 8-shapes (see MNIST dataset) and touching circles. 0- and 8- shapes can be described as a simply-connected contour relying either on a ridge detector (which will provide a midline, ignoring the width) or on their outer edge. Touching circles, like 8-shapes, can be described by a simply-connected sequence of points either like drawing a flower with two petals or like drawing an 8 with a self intersection, and then traversed either clockwise or counterclockwise. We have encountered this in currently unpublished biological data (see response to Reviewer MfnJ) and can confirm that our method is able to describe such structures successfully.
>
> **If the method assumes that shapes are simply connected, how often this is observed in evaluated data?** We have so far not encountered a case in which the simply-connected assumption breaks. We expect this to hold true especially for biological imaging data, where objects do not have holes (e.g. cells) and can be described as a ridge when self-intersecting (e.g. filaments).
>
> # Literature on rotation-invariant point networks
>
> Thank you for pointing us to related works on 3D point cloud analysis. We agree that 2D contours can also be expressed as 3D point clouds (with all contour points lying on a plane). We however identify two major differences in the problem addressed by (10.1109/TVCG.2021.3092570, 10.1007/s11263-022-01601-z, 10.1109/ICCV48922.2021.01591) and our method.
>
> **1. Supervised versus self-supervised learning:** While (10.1109/TVCG.2021.3092570, 10.1007/s11263-022-01601-z, 10.1109/ICCV48922.2021.01591) present invariant architectures that can be trained for classification, segmentation or shape retrieval in a supervised fashion (using ground truth labels), we consider the problem of learning a shape representation purely from contour data without any labels. This distinction (see response to Reviewer MfnJ) is essential when it comes to biological data exploration, which is the primary motivation for our work.
>
> **2. Point clouds versus contours:** Processing point clouds (in 2D or 3D) differs from processing contours, defined in our case as ordered sequences of points. Point clouds are in contrast unordered. Using an ordered sequence is critical for us as it allows maintaining a fixed neighborhood structure and straightforwardly reconstructing outlines for visualisation, even when parts of the contour come into close proximity or if the contour self-intersects.
>
> While we believe that a direct comparison with rotation-invariant 2D point networks is not warranted for the reasons above, we do agree that it would be valuable to integrate this discussion in our related work section and will do so in the final version of the paper if accepted.
>
> # Literature on shape descriptor methods
> Thank you for pointing us to the line of work explored in (10.1145/571647.571648). Despite its focus on 3D polygonal models and not 2D contours, similar concepts could be potentially applied in our case in future works. We propose to add this reference and discuss it in the “Statistics-based methods” paragraph of our related work section in the final version of the paper.
>
> # Other remarks
> * Following your suggestion, we will add new tables with accuracy, precision, and recall for all experiments in the supplementary materials of the final version if accepted. The results provided in our response to Reviewer HCmb already include these additional metrics.
>
> * We will increase the font size to make the legend of Figures 3 and 4 easier to read in the final version if accepted.

---

> > ### Comment · Reviewer_C6GA · 2025-04-07
> >
> > I very much appreciate significant effort that authors put into rebuttal comments for me and other reviewers. I agree with authors that method shows very strong performance on biological data and might be of potential interest for the community.
> >
> > However, there are important remaining concerns that prevent me from rising my rating:
> >
> > - I think that the paper title is too broadly positioned. To me, the current title implies more general method than one being proposed. To me, the proposed method seems to be well-design solution that tackles problem of analysis of simply-connected 2D silhouettes of biological objects (i.e. cells). Current title "ShapeEmbed" implies more general approach for shape quantification and should be compared to more general shape analysis approaches (e.g. PointTransformers, PointNets, DGCNN etc). All of these approaches are inherently applicable to 2D shapes. For example, PointNet (2016 paper) runs experiments on MNIST as well.  If comparison to more general methods is not included, I think that the title should be something like  "2DShapeEmbed: a self-supervised learning framework for 2D silhouette quantification" to reflect this particular nature of the method. Can authors comment on that?
> >
> > - Authors addressed my theoretical concern about method being applicable to only simply connected shapes. However, it is not clear whether proposed heuristics are implemented for the current iteration of the method. Can authors clarify whether they are currently used or not?

---

> > > ### Author Response · Authors · 2025-04-07
> > >
> > > ## The paper title is too broadly positioned and the current title implies a more general method than being proposed
> > > To clarify the scope of our method and taking into account your suggestion, we propose to revise the title to “ShapeEmbed: a self supervised learning framework for 2D contour quantification” in the final version. We consider this modification to be an appropriate consensus that preserves our original method’s name (ShapeEmbed) while clarifying its applicability to 2D contours. We prefer the term “contour” over “silhouette” as the latter is often understood as a “filled contour” (akin to a mask) and our approach really only relies on the points at the border of the object.
> > >
> > > ## It is unclear whether the proposed heuristics are implemented in the current version of the method
> > > We are not sure if “heuristics” here refer to the pre-processing steps required to extract an ordered sequence of points from self-intersecting object outlines, or to the processing (encoding and decoding) of these ordered sequences of points. As far as the processing is concerned, no heuristics need to be implemented - as long as an object outline is provided as an ordered sequence of points, it can be encoded and decoded successfully with ShapeEmbed regardless of whether it self-intersects or not. As mentioned in our response to reviewer MfnJ, we have observed this on biological imaging data that present such patterns but that we are not currently at liberty to disclose.
> > >
> > > If your questions concerns _how_ one can extract ordered sequences of points from masks of self-intersecting objects, this can be achieved for relatively simple datasets relying on classical image processing operations as demonstrated for instance for MNIST in this blog post: https://edwin-de-jong.github.io/blog/mnist-sequence-data/. More refined solutions have been proposed in the context of biological imaging, where complex object (self-)intersections may occur, see for instance https://doi.org/10.1007/978-3-642-15711-0_79. In our codebase, our pre-processing step is a standalone module, that isn’t part of the ShapeEmbed model per se (since the model takes as an input distance matrices, not masks), and that implements a simple contour extraction step relying on classical Python libraries. It can easily be modified or replaced by more refined and dataset-specific operations such as the two examples provided earlier.
> > >
> > > In summary, we do not claim that ShapeEmbed solves the problem of extracting ordered sequences of points from outlines _in general_, but can confirm that, provided with outlines described as ordered sequence of points, our method appropriately handles self-intersecting objects. We propose to make this point clearer with an additional sentence in Section 3.1 of the revised manuscript.

---

### Official Review · Reviewer_2G1d · 2025-03-17

**Overall Recommendation:** 4

**Summary:**

The paper proposes a novel self-supervised framework for shape embedding, which is invariant to translation, scale, and outline pixel-indexing. The learned shape representation is used in a classification task and outperforms all previous works.
## update after rebuttal
My final rating is accept.

**Claims And Evidence:**

N/A

**Essential References Not Discussed:**

N/A

**Experimental Designs Or Analyses:**

N/A

**Methods And Evaluation Criteria:**

N/A

**Other Comments Or Suggestions:**

N/A

**Other Strengths And Weaknesses:**

Strength
1. The proposed method is novel and insightful.
2. The motivation and underlying principles of the proposed method are clearly explained.
3. The experiments are solid and can support the claims of the work.

**Questions For Authors:**

N/A

**Relation To Broader Scientific Literature:**

N/A

**Theoretical Claims:**

N/A

---

> ### Author Rebuttal · Authors · 2025-04-01
>
> Thank you for your positive comments on the strengths of ShapeEmbed and for your appreciation of the way we describe and evaluate the method.

---

### Official Review · Reviewer_HCmb · 2025-03-18

**Overall Recommendation:** 3

**Summary:**

This work presents ShapeEmbed, a new approach for representation learning of 2D shapes (represented as contours/outlines). The main desiderata for such a representation are invariance to translation, rotation, scaling, reflection, and indexing.

The key ideas behind the design of this approach build upon using the distance matrix representation of a 2D shape outline as input to a convolutional neural net-based VAE. Distance matrices can be easily made invariant to everything other than indexing (what point we start from when we encode the curve). The main observation is that there is an equivalence between indexing invariance and translation invariance if the distance matrix is treated as an image. To take advantage of this, the authors implement circular padding in all convolution and pooling operations of the ResNet18 encoder, and propose a variety of reconstruction loss functions to achieve indexing invariance and regularize the training.

Comparisons are made against two classical algorithms and the state-of-the-art O2VAE on the MPEG7 and MNIST shape datasets, and on biological datasets MEF and BBBC010, by training logistic regression on top of the learned latent spaces. ShapeEmbed has a clear performance advantage.


** Post-rebuttal update **
After the rebuttal, reviewing the responses and other reviews, I have decided to maintain my initial rating.

**Claims And Evidence:**

The main claims are
1. Superior performance for shape quantification over a range of natural and biological images.
- this is demonstrated well by the strong performance over the recent O2VAE
2. Capturing variability both across and within experimental conditions in biological images
- this is qualitatively shown using a tSNE-based figure and quantitative results in the main text and the supplement
3. Indexing invariance matters and results in significant performance improvements
- Table 4 clearly shows that removing indexing invariance (no circular padding, no indexing invariance loss) significantly reduces performance.

Based on this, I conclude that these claims are supported by clear and convincing evidence. Given that the biological task of dead vs. alive C. elegans, is a bit simple (curved vs straight shapes), the significance might be limited.

**Essential References Not Discussed:**

Essential references that should be included are self-supervised computer vision algorithms like MAE, SimCLR, MoCo, DINO etc. These algorithms are also important baselines that are excluded.

**Experimental Designs Or Analyses:**

The main experimental premise is simple: Four datasets and four methods. Each method, including the proposed one, produces a feature descriptor without using any class labels. Then, a logistic regression classifier is trained on these features following 5-fold cross-validation.

The ablations cover evaluating the effect of index invariance, rotation, and translation and the effect of learning an encoding vs using distance matrices directly.

**Methods And Evaluation Criteria:**

The chosen classical baselines and SOTA O2VAE make sense. Having two "natural" datasets and two biological datasets is sufficient for showing the general applicability of the approach.

**Other Comments Or Suggestions:**

The table captions should be revised. For example, Table 1 is described as having biological imaging data, while it shows results on MNIST and MPEG-7

**Other Strengths And Weaknesses:**

Strengths:
- The work is well presented, motivated and well executed. The proposed approach has good performance

Weaknesses:
- Significance: the application domain appears to be quite limited. It is not clear why it is important to learn how to encode 2D silhouettes of objects in a self-supervised way, and the impact of driving this innovation forward. The intro can be revised to include this information.
- Significance: there have been significant efforts in self-supervised learning in computer vision through a variety of contrastive learning techniques like MAE, SimCLR, MoCo, DINO etc. These are not discussed at all, and can potentially serve as powerful baselines -- just because they do not target silhouette images does not make them not applicable to this task.
- Empirical evaluation: It would be helpful to include the biological datasets in Table 3 of the supplement to ensure that the VAE encoding makes a significant difference over just the distance matrix.

**Questions For Authors:**

My main concerns are with the clear communication of the significance of self-supervised learning of 2D silhouettes and why the prevailing self-supervised learning techniques from computer vision have not been included at all both in the discussion and experiments. The work is well executed, but the lack of reference or comparison to methods like MAE or SimCLR makes my current rating a significantly borderline accept.

If the authors can provide clarification on the significance of investigating self-supervised learning of 2D silhouettes, and why a lot of mainstream computer vision self-supervised learning algorithms are not discussed, I will be more confident in my positive weak accept rating.

**Relation To Broader Scientific Literature:**

The key contribution of the paper is figuring out how to use the distance matrix representation to add additional indexing invariance for encoding 2D shapes represented as contours. The paper empirically shows that adding this indexing invariance makes a positive impact. The main learning-based baseline O2VAE implements invariance less explicitly and has weaker performance.

**Theoretical Claims:**

This work does not make theoretical claims, but I thoroughly checked the motivation for how to build out indexing invariance and the equivalence to translation invariance makes sense.

---

> ### Author Rebuttal · Authors · 2025-04-01
>
> # Significance of investigating self-supervised learning of 2D silhouettes
>
> We refer to our reply to Reviewer MfnJ, where we clarify how we envision the method to be used in biological imaging.
>
> # Comparison with prevailing computer vision self-supervised learning algorithms
>
> We hereafter provide clarifications on how ShapeEmbed compares against a ViT model (MAE) and a contrastive learning framework (SimCLR) trained on binary masks as input (as we did for the other methods we compare against).
> * Masked AutoEncoders (MAE). We benchmarked against the 3 “off-the-shelf” MAE ViT (https://github.com/facebookresearch/mae) configurations (“base”, “large”, and “huge”). We resized the masks to 224x224, the input size expected by MAE by default. We used a batch size of 16 (as in the original MAE paper) and 200 epochs (as ShapeEmbed).
> * SimCLR. We created a SimCLR model with a ResNet18 backbone with 128 output dimensions relying on the original codebase (https://github.com/sthalles/SimCLR/). We trained for 200 epochs (as ShapeEmbed) using the default configuration and set of transforms to create positive pairs.
>
> ## MNIST
> |Method|acc|prec|recall|F1|Log loss|
> |:-:|:-:|:-:|:-:|:-:|:-:|
> |mae_vit_base|0.953±0.003|0.953±0.003|0.952±0.004|0.953±0.026|0.062±0.020|
> |mae_vit_large|0.840±0.009|0.841±0.009|0.840±0.009|0.840±0.009|0.520±0.021|
> |mae_vit_huge|0.921±0.004|0.919±0.003|0.921±0.004|0.923±0.005|0.369±0.018|
> |simCLR_rs18|0.598±0.011|0.594±0.012|0.598±0.011|0.593±0.011|1.188±0.033|
> |**ShapeEmbed**|**0.963 ± 0.005**|**0.963 ± 0.005**|**0.963 ± 0.005**|**0.963 ± 0.007**|**0.187 ± 0.020**|
>
> ## MPEG-7
> |Model|acc|prec|recall|F1 score|Log loss|
> |-|:-:|:-:|:-:|:-:|:-:|
> |mae_vit_base|0.675±0.024|0.660±0.016|0.675±0.024|0.646±0.001|1.471±0.071|
> |mae_vit_large|0.654±0.037|0.637±0.037|0.654±0.037|0.627±0.040|1.465±0.112|
> |mae_vit_huge|0.633±0.166|0.615±0.001|0.601±0.045|0.600±0.010|1.767±0.079|
> |simCLR_rs18|0.141±0.016|0.145±0.022|0.141±0.016|0.128±0.020|3.502±0.522|
> |**ShapeEmbed**|**0.763±0.037**|**0.716±0.002**|**0.763±0.036**| **0.751±0.024**|**1.158±0.206**|
>
> ## MEF
> |Method|acc|prec|recall|F1|Log loss|
> |-|:-:|:-:|:-:|:-:|:-:|
> |mae_vit_base|0.537±0.031|0.539±0.030|0.546±0.029|0.537±0.030| 0.895±0.024|
> |mae_vit_large|0.535±0.019|0.534±0.020|0.535±0.018|0.532±0.019 |0.885±0.028|
> |mae_vit_huge|0.549±0.023|0.552±0.024|0.549±0.023|0.549±0.023| 0.830±0.034|
> |simCLR_rs18|0.444±0.0292|0.451±0.032|0.444±0.0292|0.434±0.0316| 1.019±0.0203|
> |**ShapeEmbed**|**0.745±0.006**|**0.751±0.006**|**0.745±0.005**|**0.746±0.006**|**0.640±0.016**|
>
> ## BBBC010
> |Model|acc|prec|recall|F1 score|Log loss|
> |-|:-:|:-:|:-:|:-:|:-:|
> |mae_vit_base|0.628±0.105|0.633±0.107|0.6285±0.105|0.597±0.119|0.7161±0.156|
> |mae_vit_large|0.720±0.580|0.723±0.058|0.721±0.060|0.514±0.072|0.632±0.082|
> |mae_vit_huge|0.657±0.081|0.671±0.090|0.657±0.081|0.718±0.059|0.649±0.083|
> |simCLR_rs18|0.567±0.115|0.569±0.119|0.567±0.115|0.562±0.117|0.762±0.125|
> |**ShapeEmbed**|**0.872±0.015**|**0.866±0.009**|**0.866±0.009**|**0.866±0.008**|**0.509±0.136**|
>
> We hypothesize that the sub-par performance of MAE is due to ViT not being suited to the problem we study: we seek to encode the shape of individual objects in small images coming from small datasets (by ViT standards). We believe that the poor performance of SimCLR is due to the construction of positive pairs, here defined as an image and an augmented version of itself. To achieve rotation, scaling, and positional invariance with SimCLR, we would need to define an alternative way of creating positive pairs of masks that would comprehensively cover all the transformations we normalize for in ShapeEmbed, which is beyond the scope of a direct comparison. We are interested in exploring ways to use a contrastive framework on a distance matrix input in future works, but doing so requires further investigation and is out of scope for this paper.
>
> We did not manage to benchmark against MoCo and DINO in the rebuttal period, but anticipate that we would reach similar conclusions as the challenge of using contrastive learning and ViT models for our task would apply as well.
>
> If accepted, we will add the following results to Tables 1 and 4, and include the discussion in a new section of our supplementary material.
>
> # Other remarks
>
> * Following your suggestion, we will include the new results below on the two biological datasets (BBBC010 and MEFs) in Supplementary Table 3 in the final version if accepted.
>
> Dataset|Distance&nbsp;matrices|||||ShapeEmbed|||||
> |-|:-:|:-:|:-:|:-:|:-:|:-:|:-:|:-:|:-:|:-:|
> ||acc|prec|recall|f1|log|acc|prec|recall|f1|log|
> |BBBC010|0.737±0.025|0.737±0.025|0.734±0.026|0.737±0.026|2.889±0.686|**0.872±0.015**|**0.866±0.009**|**0.866±0.009**|**0.866±0.008**|**0.509±0.136**|
> |MEF|0.343±0.007|0.452±0.024|0.343±0.007|0.299±0.006|1.202±0.066|**0.745±0.006**|**0.751±0.006**|**0.745±0.005**|**0.746±0.006**|**0.640±0.016**|
>
> * Thanks for flagging the typo in the caption of Table 1, we will revise it in the final version.

---

> > ### Comment · Reviewer_HCmb · 2025-04-09
> >
> > Thank you for the effort in performing these additional comparisons, I believe they will be a valuable addition to this paper.

---

### Decision · Program_Chairs · 2025-05-01

**Decision:**

Reject

**Comment:**

The submission got 2 negative and 2 positive recommendations eventually. The reviewers were mainly concerned about the significance, evaluations, and the reliance on foreground masks. The authors addressed some of these concerns in the rebuttal, but not all of them. The reviewers briefly discussed their concerns in the reviewer discussion period. One reviewer and the AC were also concerned about one review that is positive but too short to meet the criterias of reviews. Hence, the AC reduced its credibility. The AC read through the submission, all reviews, rebuttals, and confidential comments to the AC. The AC agreed with most of the reviewers and believes that the authors should make a major revision accordingly. Per this, the AC made a decision of rejection. The decision was also approved by the senior AC.